# Genetic and Molecular Determinants of Lymphatic Malformations: Potential Targets for Therapy

**DOI:** 10.3390/jdb10010011

**Published:** 2022-02-08

**Authors:** Su Yeon Lee, Emma Grace Loll, Abd-Elrahman Said Hassan, Mingyu Cheng, Aijun Wang, Diana Lee Farmer

**Affiliations:** 1Division of Pediatric, Thoracic and Fetal Surgery, University of California Davis Medical Center, Sacramento, CA 95817, USA; ashas@ucdavis.edu; 2Center for Surgical Bioengineering, Department of Surgery, University of California Davis, Sacramento, CA 95817, USA; egloll@ucdavis.edu (E.G.L.); aawang@ucdavis.edu (A.W.); 3Department of Pathology and Laboratory Medicine, University of California Davis Medical Center, Sacramento, CA 95817, USA; mngcheng@ucdavis.edu

**Keywords:** lymphatic malformation, cystic hygroma, PI3CKA

## Abstract

Lymphatic malformations are fluid-filled congenital defects of lymphatic channels occurring in 1 in 6000 to 16,000 patients. There are various types, and they often exist in conjunction with other congenital anomalies and vascular malformations. Great strides have been made in understanding these malformations in recent years. This review summarize known molecular and embryological precursors for lymphangiogenesis. Gene mutations and dysregulations implicated in pathogenesis of lymphatic malformations are discussed. Finally, we touch on current and developing therapies with special attention on targeted biotherapeutics.

## 1. Introduction

The lymphatic system works to maintain extracellular fluid homeostasis, aid in fat absorption and transportation, and optimize immunity. Lymphatic malformations are congenital low flow vascular malformations of the lymphatic vessels. They can be differentiated into three general categories: (1) abnormalities of lymph vessels and nodes leading to inadequate clearance of lymph and subsequent lymphedema, (2) mass lesion composed of cysts, and (3) disordered circulation of chyle in central conducting lymphatic channels [1]. This review focuses on the genetic and molecular pathogenesis of cystic mass lesions. While commonly referred to as cystic hygroma, lymphatic anomalies have extremely varied presentations other than simple cysts and may be combined with malformations of capillaries, veins, and arteries.

The overall incidence of lymphatic malformations ranges from 1 in 6000 to 1 in 16,000, with no predilection for sex or race [2]. Seventy-five percent of the cases occur primarily in the lymphatic-rich head and neck region but can exist in any part of the body, including the axilla, mediastinum, and abdomen [3]. According to the International Society for the Study of Vascular Anomalies (ISSVA), lymphatic malformations are classified by size, with discrete cysts more than 1–2 cm referred to as macrocystic, and those smaller than 1–2 cm as microcystic [4]. The lesions can vary from focal and well-defined cystic masses to infiltrative and poorly defined multicystic lesions. On histology, they are seen to be lined by single layer of flattened endothelial cells with lymphatic phenotype staining with prospero homeobox protein 1 (Prox1) and podoplanin D2-40. For clinical diagnosis, the hematoxylin and eosin-stained histology is sufficient. They also have abnormally formed smooth muscle resulting in fibrovascular vessel walls of varying thickness (Figure 1) [5]. Their lumens are filled with eosinophilic proteinaceous lymph fluid, which allows it to be easily seen on imaging. With advancements in prenatal care and imaging, lymphatic malformations are being more readily classified and diagnosed on prenatal ultrasound. Respiratory distress, recurrent infections, and deformity are the main indications for treatment. Treatment is based on surgical excision, although sclerotherapy and pharmacologic treatments, including use of immunosuppressive agents, are being studied.

## 2. Lymphatic Development

While the exact embryonic origin of these malformations is unclear, they are believed to be the result of developmental defects of dilated lymphatic channels [6]. There are multiple theories on lymphatic development. Initially proposed by Sabin, in the “centrifugal theory,” primordial lymphatic sacs were thought to sprout from developing central veins and propagate peripherally [7]. This model was further supported by experiments from Lewis that demonstrated detachment of portions of sub-cardinal vein from the venous system to form lymphatic ducts in rabbits [8]. McClure proposed a different theory in which lymphatic vessels form by coalescing isolated spaces in the mesenchyme, which subsequently transform into lymphatic endothelial cells (LECs) [9]. These LECs develop into a primitive lymphatic network and connect to the venous system in a “centripetal” model in which the lymphatic system develops peripherally to centrally.

Recent work in different model organisms has demonstrated variation in the origin of LECs. Avian and xenopus models have demonstrated dual origin of venous endothelial cells and mesenchymal lymphangioblasts, combining both theories [10,11]. Live imaging zebrafish embryos allowed in vivo tracking of LEC progenitors originating in the posterior cardinal vein and migrating dorsally to the thoracic duct, thus supporting Sabin’s centrifugal theory [12]. Okuda et al. demonstrated that in zebrafish, the facial lymphatic network develops through sprouting from the common cardinal vein and additional populations of lymphangioblasts recruited to this main sprout [13]. However, the formation of lymphatic networks differs between mammals and zebrafish, with lymphangiogenesis and angiogenesis temporally separated; in mice, lymphatic structure forms after lymphatic specification following angiogenesis, while in zebrafish, venous sprouts give rise to both intersegmental veins and lymphatic precursors [14]. As such, prox1 signaling to specify LEC in mouse is integral for lymphangiogenesis, but dispensable in zebrafish. Additionally, lineage-tracing studies in mice demonstrated that LECs were derived from venous origins. Particularly, in hematopoietic deficient mice, LECs from venous-derived lymph sacs proliferated into lymphatic vasculature; hematopoietic cells did not contribute to development of the lymphatic system [15].

Adding to this complexity, organ-specific lymphatic vasculature pools progenitor cells from different populations. In the murine model, skin, cervical, and thoracic lymphatic vessels were formed from Tie2 lineage venous derived LEC progenitors, while most lumbar and cardiac lymphatic vessels were formed by coalescence of isolated Tie2 negative non-venous LEC progenitors [16,17]. Mesenteric lymphatic vessels were seen to originate from two distinct sources: Tie2 positive venous endothelial cells and PDGFB and c-Kit positive hemogenic endothelial cells [18]. Lymph nodes have been suggested to arise from nestin positive precursors [19]. However, the results of these experiments with lineage tracing in mice should be interpreted with caution. The efficiency of Cre-mediated recombination may vary, therefore introducing a chance for misinterpretation of biological heterogeneity as incomplete recombination. Additionally, there may be other markers that are more specific for lymphatic lineage and able to mark cells earlier in development.

An unknown signal currently triggers polarized expression of homeobox gene Prox1 in a subpopulation of endothelial cells, committing them to lymphatic lineage and promoting budding [20]. Underexpression or overexpression of Prox1 in mice were lethal secondary to lack of lymphovascular development and edema from increased permeability, respectively [20,21]. Prox1 expression in mice occurs at E9.5 in subpopulation in cardinal vein until E14.5, and is maintained by nuclear hormone receptor Coup-TFII and SOX18 [22]. SOX18, SRY-related HMG domain family of transcription factors that bind to Prox1 proximal promoter, is integral for early maintenance of Prox1 but is not past E14.5 during lymphangiogenesis [23]. Lymphatic vessel endothelial hyaluronan receptor 1 (LYVE1), podoplanin, and vascular endothelial growth factor receptor 3 (VEGFR3) are upregulated by these PROX1 positive progenitors [24,25,26].

Early LECs start producing secondary lymphoid chemokines such as vascular endothelial growth factor c (VEGFC) and increase expression of VEGFR3 [27]. VEGFC binds to VEGFR3, mediated by neuropilin 2 (NRP2), then activates phosphoinositide 3-kinase (PI3K)/AKT and RAF1/MEK/ERK signaling pathways, which induces LEC migration and SOX18 expression, respectively [28,29,30,31]. VEGFC is integral for initial lymphatic development, as seen by the lack of lymph vessels sprouting from committed endothelial cells in VEGFC null mice [32]. VEGFC and VEGFD, which also activates VEGFR3, need to be activated to their mature form by proteolytic cleavage [33,34,35]. Of note, VEGFD is not required for lymphangiogenesis, unlike VEGFC [36]. Collagen- and calcium-binding epidermal growth factor domains 1 (CCBE1) enhances the cleavage of VEGFC by A disintegrin and metalloprotease with thrombospondin motifs 3 (ADAMTS3) [37]. CCBE1 knock-out mice have shown similar phenotype as VEGFC null mice, with specified lymphatic progenitors failing to migrate from cardinal vein [38]. Finally, angiopoietin 2 (ANGPT2) and its tyrosine kinase receptor TIE2 are involved with maturation and patterning of these lymphatic vessels continuing postnatally [39]. Of note, small populations of Prox1 positive LECs remaining in the veins form lymph-venous valves [40].

First, morphologically visible primordial lymph sacs in humans are seen at 6 to 7 weeks gestational age, consisting of paired jugulo-axillary (which gives rise to lymph vessels of head, neck and arms) and lumbo-iliac (gives rise to lower half of the body) lymph sacs, and singular retroperitoneal lymph sac and chylocyst [41]. Starting at weeks 6.5 to 7, endothelial buds (lymphatic primordia) develop in the jugular region after arterial and venous development. These buds unite to form plexuses which further develop into paired jugular sacs and axillary lymph sacs [42]. By 8 weeks, both axillary and jugular sacs enlarge and form single communication to the internal jugular vein and to each other on either side. Jugulo-axillary lymph sacs continue to grow through week 9 predominantly in cranial and dorsolateral directions to eventually reach the subcutaneous area of the posterior neck compartment. Rudimentary bilateral thoracic ducts are seen to anastomose with left jugulo-axillary lymph sac and lumbar lymph plexuses. Lymphatic valves are starting to be formed at this time. All lymphatic primordia fuse and one continuous system is formed at about 10 weeks gestational age. Continuous peripheral development from these central primordial occurs in all trunks, and these primordial lymph sacs become less prominent as lymphatic branches continue to develop.

In the end, the lymphatic system develops into a network of thin, permeable, blind-ended capillaries that drain into larger collecting vessels complete with valves. Lymphatic capillaries consist of a single layer of LECs that lack a continuous basement membrane [43]. Initial lymphatics have discontinuous button-like junctions between endothelial cells that allow for easy uptake of interstitial fluid and macromolecules [44]. Lymph, after uptake by initial lymphatics, flows through collecting lymphatic vessels characterized by zipper-like junctions [44]. Valves allow for unidirectional transport of lymph through these collecting vessels, into lymph nodes to larger collecting ducts and back to venous circulation. Collecting vessels and onwards have basement membrane and are surrounded by pericytes and smooth muscle cells. Additional skeletal muscle contractions, arterial pulsation, and gravity provide propulsive force for lymph transport.

## 3. Pathogenesis

Few major theories on cystic lymphatic malformation pathogenesis have been proposed (Figure 2). One theory assumes McClure’s centripetal theory of mesenchymal origin of lymphangiogenesis and suggests primordial lymphatic sacs fail to fuse with venous system [45]. This results in isolated lymphatic canals that dilate and form cystic malformations. Zadvinskis et al. and van der Putte both demonstrated cystic lesions in the neck have missing lympho-venous connections in jugulo-axillary and jugulo-subclavian areas [42,46]. Others have assumed that tissue abnormally sequestered early in embryogenesis was the culprit. These fail to join normal central lymphatic channels and become malformations [47]. Abnormal budding of lymphatic structures may be another etiology of lymphatic malformations. Aberrant buds lose connections with lymphatic primordia and canalize to form lymph filled cysts. These structures branch and grow in a disorderly manner, forming cysts without lymphatic drainage [41]. More recently, efforts are being taken to elucidate molecular mechanisms behind this aberrant growth.

### 3.1. Clinical Syndromes

While lymphatic malformations can exist as solitary lesions, many of them are associated with specific syndromes and genetic mutations. Generalized lymphatic anomaly (GLA) and Gorham–Stout disease (GSD) are complicated lymphatic malformations that involve multiple body sites.

GLA most often appears in childhood with proliferation of normal mature lymphatic ducts and the formation of numerous cysts in any organ of the body. While the cysts themselves are benign, the size, critical locations, and secondary infections can cause significant morbidity and mortality [48]. Kaposiform lymphangiomatosis has been recently recognized as a subtype of GLA, with foci of spindle endothelial cells amid a background of malformed lymphatic channels [49]. It affects multiple organs, with thoracic involvement being most common. Extrathoracic disease can manifest in bone including in the extremities, and in abdominal viscera. Mortality is high with cardio-respiratory failure from thoracic disease, with only 34% overall survival, and mean interval between diagnosis and death of less than 3 years [50].

GSD is a very rare disorder characterized by proliferation of capillary-sized vascular channels, including lymphatic vessels, in bone, and nearby soft tissue resulting in progressive osteolysis. While GSD can develop in any musculoskeletal site, it most commonly occurs in the shoulder and the pelvic girdle. Skull, humerus, sternum, ribs, femur, and spine have all been seen to be affected [51,52]. Morbidity and mortality is high secondary to osteomyelitis, spinal cord involvement including paraplegia from vertebral lesions, and chylothorax from extension into pleural cavity or thoracic duct [52].

Additionally, there are mixed venous, arterial, and lymphatic vascular malformations that are associated with other anomalies. These include Klippel–Trenaunay syndrome (KTS) with capillary, venous, and lymphatic malformation and limb overgrowth, CLOVES (congenital lipomatous overgrowth, vascular malformations, epidermal nevis, spinal/skeletal anomalies/scoliosis) syndrome with capillary, venous, lymphatic, and arterio-venous malformations and lipomatous overgrowth, and CLAPO (capillary vascular malformation of the lower lip, lymphatic malformations of the head and neck, asymmetry and partial or generalized overgrowth) syndrome with lower lip capillary malformation, cervical lymphatic malformation, and asymmetrical somatic overgrowth as part of PIK3CA-related overgrowth spectrum (PROS) which results from somatic PIK3CA activation mutation. Additionally, Proteus syndrome has capillary, venous, and lymphatic malformations, and asymmetrical somatic overgrowth from AKT1 mutation.

### 3.2. Genetic Mutations

In certain cases of lymphatic malformation, its cause can be easily determined if it is a feature of a greater condition like KTS or CLOVES syndrome in PROS. However, pinning down the genetic source of solitary malformations has proven challenging as there is no historically known cause. Research in the past decade has helped elucidate various genes that might be involved in the development of lymphatic malformations and whether they are isolated or a part of a greater medical condition.

One such gene of interest is the PIK3CA gene, which encodes for the catalytic subunit of PI3K. The PI3K/AKT pathway is integral in lymphatic development by inducing LEC migration. As previously mentioned, PIK3CA somatic mosaic activating mutation, particularly in the catalytic p110α subunit, is seen in a number of mixed vascular malformations [53,54]. Using samples of affected tissue from Boston Children’s and Seattle Children’s hospitals, Luk et al.’s analysis revealed five activating mutations in the gene that were present at low frequencies (<10%) in patients with isolated lymphatic malformations, KTS, CLOVES, and fibro-adipose vascular anomaly (FAVA) [55]. These five mutations accounted for ~80% of cases and considering both the somatic mosaic nature of their results and proliferative nature of the PI3K pathway, this may reflect a biological mechanism where mutant cells recruit wild-type cells over the course of the overgrowth process [55,56]. Recent studies from Le Cras et al. demonstrate that LECs from patients with capillary lymphatic venous malformations with PIK3CA mutations recapitulate the patient’s lesion when injected into immunocompromised mice in a xenograft model [54]. Additionally, there may be a mutation in PIK3CD that acts in the PI3K pathway in conjunction with PIK3CA. Wang et al. demonstrated novel PIK3CD mutation in human samples of lymphatic malformation, and in vitro studies in human umbilical vein endothelial cells with these mutations demonstrated increased cell proliferation and hyperactivation of mTOR pathway [57].

AKT1 is another gene in the PI3K/AKT signaling pathway that has been seen with activating mutation associated with Proteus syndrome. As previously discussed AKT stimulates LEC migration during lymphangiogenesis. Lindhurst et al. performed DNA exome sequencing of varying samples from 29 patients with Proteus syndrome and found that 89% of tested patient samples had somatic activating mutation in the AKT1 gene [58]. Importantly, they demonstrated somatic mosaicism ranging from 1–50% of mutant alleles in different tissues and cell lines of patients with varying levels of phosphorylation. At the current time, it is unclear during which stage of development the somatic mutation occurs in embryo. Additionally, there is a wide variability in lymphatic presentation of Proteus syndrome, and there is no clear association between proportion of mutant alleles to lymphatic malformation phenotype. However, there have been associations between the frequency of cutaneous manifestations, including lymphatic malformations, with clinical severity of proteus syndrome and extra-cutaneous manifestations [59]. An additional case report describe novel somatic mosaic heterozygous duplication encompassing exon 3 to 15 of AKT1 gene in a patient with large left sided cervical cystic lymphatic malformation [60].

Similarly, the Burrows et al. case study of a patient with Parkes–Weber syndrome (PWS) uncovered the RASA1 gene as a possible genetic source for the phenotypic expression of lymphedema. In adult mice studies, the gene was revealed to be an integral part in regulating the Ras signaling pathway, as its overactivity leads to hyperplasia of the lymphatic vasculature [61]. Utilizing whole-exome sequencing and Sanger sequencing, they confirmed a frameshift mutation in RASA1 in the patient and his father. Additionally, use of investigational near-infrared fluorescence lymphatic imaging (NIRFLI) in concert with radiographic lymphangiography further affirmed the association between RASA1 and lymphatic abnormalities. Aberrant lymphatics were observed in both the PWS patient and Rasa1 knockout mice, suggesting the mutation in RASA1 and subsequent irregular Ras activity could explain such lymphatic vessel anomalies [56]. While these studies successfully unveil possible genes for further investigation, it has yet to be determined why such mutations create an isolated malformation in one individual as compared to the presence of a malformation as part of a greater syndrome in others.

Broadly, the findings presented from these various studies do not point to a definitive cause of lymphatic malformations, but rather have revealed new genetic targets to explore which can be used to investigate more effective treatments for all types of lymphatic malformations. In addition, many of these mutations including ones in the PI3K pathway are significant contributors in vascular malformations and have been identified as potential targets for pharmaceutical therapies [62]. Although these may acquire mutations at different stages of development, there are clear parallels between lymphangiogenesis and angiogenesis. This lends further evidence to using these genes as potential targets for future therapies for lymphatic malformations as well.

### 3.3. Gene Dysregulation

Gene dysregulation can also play a role in pathogenesis. Another gene being closely studied is VEGFC. In a PIK3CA driven microcystic lymphatic malformation mouse model, VEGFC/VEGFR3 signaling was integral in the growth of the lesion [63]. Furthermore, Han et al. investigated the relationship between Hypoxia-inducible factor-1α (HIF-1α) and VEGF receptor 3 (VEGFR-3) using samples obtained from 20 patients with microcystic, macrocystic, or mixed lymphatic malformation [64]. In brief, HIF-1α binds to HIF-1β under hypoxic conditions to trigger the HIF-1 cascade to mediate cells and their metabolism while VEGFR3 is localized to LECs and associated with lymphangiogenesis. The results indicate that both HIF-1α and VEGFR3 were upregulated in patient samples compared with normal tissues. Additionally, Han and collaborators engineered a stable, HIF-1α-overexpressing human LEC (HLEC) cell line which had increased cell migration and colony formation of HIF-1α-overexpression cells compared to control HLECs [64]. Previous studies have shown increased expression of VEGFR3 and VEGFC in various cancer cell lines under a hypoxic environment [65,66,67], and combined with the results from Han et al., suggest that the genes encoding for HIF-1α and VEGFR3 could also be involved in the malformation of lymphatic vessels leading to lymphatic malformation.

Gomez–Acevedo et al. analyzed tissue samples from 18 pediatric patients—six microcystic, seven macrocystic, and five normal controls—to determine the underlying genetic and physiologic differences between groups. Through transcriptome analysis, they discovered 426 differentially expressed genes of which 192 and 234 were upregulated in microcystic and macrocystic samples, respectively [68]. Among the overexpressed genes in microcystic samples is eukaryotic initiation factor 4A-1 (EIF4A1), which encodes for the RNA helicase subunit of the eukaryotic initiation factor 4F (eIF4F) complex [69]. As a result, proteins such as c-Myc and VEGF are produced, and in turn, promote cell survival, proliferation, and angiogenesis which could account for the abnormalities in lymph vessels. Additionally, the upregulation of metastatic suppressor NME1 points to limitations in cell spreading and motility and may explain the condensed, fibrous morphology of microcystic lymphatic malformations [68]. While the microcystic-related genes demonstrate proliferative mechanisms and imitate oncogenic processes, the upregulated genes for macrocystic samples are related to cell differentiation, embryogenesis, adipogenesis, cell adhesion, and adaptation to hypoxia. Upregulation of microfibril-associated glycoprotein 2 (MFAP2), VEGFB, and angiopoietin-like 2 (ANGPTL2) denote angiogenic events related to cell sprouting and sprout formation [68]. Particularly of interest is the abnormal expression of extracellular matrix protein dermatopontin (DPT) which modulates transforming growth factor beta (TGF-β). In conjunction with an increase in TGF-β3, its upregulation indicates that macrocystic lymphatic malformations may arise due to improper regulation and pruning of developing lymphatic networks [68]. The transcriptome differences discovered in this study illuminate differences in genetic causes and mechanisms that should be further investigated to better understand the formation of microcystic and macrocystic lymphatic malformations. While these dysregulated pathways need further studies to demonstrate causality, this is a promising area of study for potential therapeutic targets.

## 4. Clinical Implications

Most lymphatic malformations persist postnatally, with spontaneous regression rates between 2.3% to 41% [70]. While some minimally symptomatic lesions can be observed, most will undergo treatment for aesthetic or functional reasons.

### 4.1. Current Surgical Treatment

Surgical excision is the mainstay of treatment for lymphatic malformation, particularly in lesions causing functional disability secondary to compression. Airway obstruction, dysphagia and problematic bleeding should be addressed promptly, and with excision. For large prenatally diagnosed cervical lymphatic malformation, ex-utero intrapartum therapy (EXIT) is indicated to secure the airway [71]. While small or macrocystic lesions in general can be readily excised, lesions of larger size, diffuse and microcystic in nature, or involving surrounding structures such as nerves or blood vessels become very complicated and result in incomplete removal. In one case series of 63 patients, 14% had recurrence of macrocystic cervicofacial lymphatic malformation after excision, and associated factors for recurrence included bilaterality and advanced staging [72]. Cosmetic appearance, impaired functionality following excision, and scarring, in addition to other potential complications, including fistulization, infection, and dehiscence, need to be kept in mind. Carbon dioxide laser and radiofrequency ablation have been used intraoperatively as an adjunct for intraoral disease [73,74].

Percutaneous treatment with injection sclerotherapy is another option for macrocystic lymphatic malformations. Introduction of sclerosing agent induces endothelial inflammation leading to secondary occlusion, fibrosis and contraction of the structure. Picibanil (OK-432), doxycycline, and bleomycin are commonly used sclerosing agents. One randomized control study demonstrated substantial response to picibanil sclerotherapy in 94% of patients with macrocystic disease and 63% in mixed macrocystic and microcystic disease [75]. Bleomycin also demonstrated similar efficacy with 80.3% of patients with macrocystic lesions and 71.4% with mixed lesions having complete response [76]. Alcohol, doxycycline, and sodium tetradecyl sulfate have also been used, but the best sclerosing agent is still unclear [77,78]. Currently, there is no established guideline for the best choice of initial therapy between surgical excision, sclerotherapy or medical treatment. A retrospective cohort study demonstrated similar effectiveness for both initial surgical excision and sclerotherapy controlling for lesion stage and type [79].

### 4.2. Current Medical Treatment

Similar to surgical treatment, there is no consensus on best pharmaceutical treatment for lymphatic malformation. Sirolimus, an immunosuppressant, is currently the most commonly used agent for treatment of lymphovascular anomalies. Sirolimus is an inhibitor of mTOR, a downstream protein kinase in the PI3K pathway. While there is a lack of randomized control trial data demonstrating its effectiveness for lymphatic malformations, a number of retrospective and nonrandomized studies describe a decrease in lesion size in majority of patients with oral sirolimus [80,81]. In particular, a large prospective study by Adams et al. demonstrated disease response for mixed capillary or venous lymphatic malformation, and favorable response to microcystic lymphatic malformations with 1 year of oral sirolimus treatment. The most common side effects of oral sirolimus are related to its immunosuppressive effects including anemia, thrombocytopenia and leukopenia [82,83]. It is difficult to draw conclusions for topical sirolimus, as only 3 studies including 7 patients total have been described [81]. There are phase 2, multi-centered trials underway looking at efficacy of topical sirolimus for treatment of microcystic lymphatic malformation (NCT05050149 and NCT03972592). Other medications that have been discussed include sildenafil and propranolol, which are thought to ameliorate symptoms though vascular smooth muscle relaxation from inhibition of cGMP breakdown and reduction of lesion growth by downregulation of VEGF expression, respectively [84,85,86]. However, both have been reported in small case studies with mixed results.

## 5. Future Therapies

As more molecular players of the disease are being identified, these can represent potential targets for new therapeutics. One of those include PIK3CA gene, which is one of the most common mutations in cancer of all types [87]. Of note, human cancer often contains multiple oncogenic mutations along the same PI3K pathway in comparison to the heterozygous state for PROS [88]. There are a number of PI3K inhibitors that are being investigated for oncologic treatment including pan-PI3K inhibitors, such as copanlisib or buparlisib, and specific p110α inhibitors, such as alpelisib [89]. Alpelisib is currently approved for use in PIK3CA-mutated, hormone receptor positive advanced breast cancer. In a murine model of PROS/CLOVES, administration of alpelisib improved the survival, decreased tissue changes, and reduced cell proliferation of affected organs of PROS/CLOVES mice compared to wild type [90]. In a different mouse model of PROS, alpelisib again improved survival of mice, but also was able to prevent development of lymphatic malformations [91]. Alpelisib has been used in small groups of patients, with progressive reduction in tumor size, improvements in secondary organ dysfunctions including cardiac and renal function, musculoskeletal deformities and general clinical status [90,91]. However, larger scale prospective, randomized controlled trials are needed to fully evaluate the efficacy of these specific inhibitors. Topical application of PI3KA inhibitor is also under investigation, with a phase 1 clinical trial currently underway (NCT04409145).

Another strategy could include inhibiting AKT, a direct downstream serine threonine kinase of PI3K with a mosaic activating mutation that results in Proteus syndrome [58]. The pan-AKT inhibitor miransertib has been used in patients with cancer [92,93]. A dosing study of miransertib in adults and children with Proteus syndrome has been completed and case series of 2 patients has shown improvements in quality of life [94,95]. A phase 1/2 study is currently underway to establish efficacy and safety of miransertib for PROS (NCT03094832).

Lastly, photoacoustic technology, based on nonradiative absorbed energy combined with targeted lymph vessel endothelium labeling is being studied for mapping and potential therapeutic ablation. Kim et al. have used bioconjugated gold based nanoparticles with antibodies to LEC marker LYVE1 as photoacoustic and photothermal contrast agents in mice to precisely map lymphatic vasculature and malformations [96]. In addition, labeling of target cells can increase therapeutic efficacy and optimize guidance of laser treatment. In the same murine model with LYVE1 antibody conjugated gold nanoparticles, there was a 6-fold increase in laser energy with highly localized damage around zones of high concentration of gold nanoparticle clusters without deleterious effects in surrounding tissue [96]. Future studies are needed to establish bioconjugated nanoparticles as a potential targeted single diagnostic and therapeutic platform for microcystic lymphatic malformations. This will allow this theragnostic versatile tool to overcome current imaging technique limitations and possibly be utilized for other lymphatic related diseases.

## 6. Conclusions

While large strides into understanding the origins of lymphatic vessels and the molecular underpinnings of lymphatic malformations have been taken, the complexity of different contributions of endothelial cells and initial signaling for lymphangiogenesis is still being teased out. Further studies and investigations into understanding molecular mechanisms contributing to the specification and growth of lymphatic vessels will open doors to many new targeted therapies, including perhaps in utero therapies. Additional studies will also enrich the efforts to treat other diseases of lymphatic system including lymphedema and cancer.

## Figures and Tables

**Figure 1 jdb-10-00011-f001:**
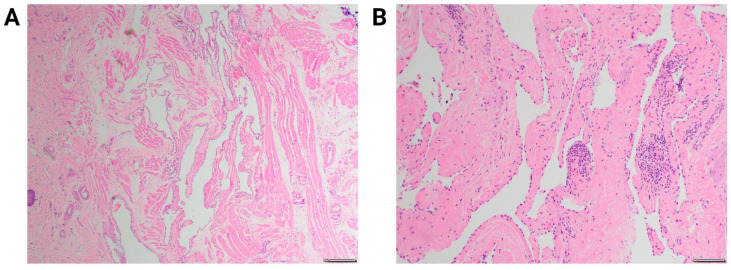
Hematoxylin and eosin stained of lymphatic malformation. (**A**) 40× image shows variably sized anastomosing vascular space lined by endothelial cells with stromal fibrosis of the malformation walls. (**B**) 100× image shows small with bland appearing endothelial cells with oval to flattened nuclei. Lymphocytes are present within both dilated lumina and septa.

**Figure 2 jdb-10-00011-f002:**
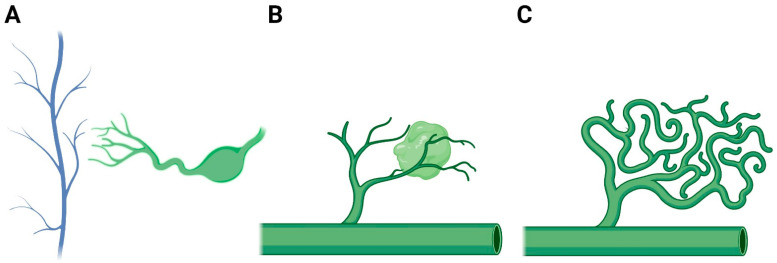
Major hypothesis regarding pathogenesis of lymphatic malformations. (**A**) McClure’s centripetal theory of mesenchymal origin of lymphangiogenesis where primordial sacs fail to fuse with venous system. (**B**) Tissue abnormally sequestered early in embryogenesis and fail to join central lymphatic channels. (**C**) Abnormal budding of lymphatic structures. Created with Biorender.com.

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
