# Peer review of "Genetic and Molecular Determinants of Lymphatic Malformations: Potential Targets for Therapy"

_jdb, 2022, doi:10.3390/jdb10010011_

Round 1

Reviewer 1 Report

This manuscript is a elegant summary of lymphatic malformations that is worthy of publication as a resource in the area. There are some adjustments suggested below to ensure clarity for readers before publication.

Major notes:

  • Please consider revising the title. This does not seem grammatically correct with respect to the use of the word “implications” in this context. “The implications of genetic and molecular alterations to the development of lymphatic malformations” might make more sense, or perhaps replacing “implications” with “underpinnings” “targets” or “determinants” eg Genetic and molecular determinants of lymphatic malformation: implications for therapy. Just suggestions, but please ensure the title reflects the core purpose of the paper.
  • The abstract needs to be rewritten. Several statement sentences are provided to give background to the disorder but we are not actually informed about what will be discussed or analysed in this review. At the end, “finally, this review will….” This final sentence is disconnected from the lack of any previous statement about what this review will demonstrate/summarise/analyse. What is the purpose of this review exactly? Will you summarise the information, or is new information synthesised/knowledge gaps identified/conclusions drawn through review of this material? Be explicit with statement of purpose.
  • Section 3.2 – according to the title of the paper this section should be a major focus. I would suggest revision of this section to ensure that identified genetic mutations vs gene dysregulation are clearly separated. Further, it is suggested that these downstream regulated genes are worth investigating, but the context must be clarified. For example, are you suggesting that they should be investigated as potential mosaic mutations in some disorders? Or suggesting they should be investigated as potential therapeutic targets? Please be explicit in your meaning of what their identification means for future endeavours in the field.
  • In relation to the above, in section 5 AKT1 is identified as a mosaic mutation in PROS. This should be discussed in section 3.2 if this is indeed a full summary of the literature with regard to genetic and molecular changes associated with this disorder and their implications for development or treatment.
  • Optional – If the focus is on postulating mutations or targets to be identified in this disorder, comparison to vascular malformations (capillary, brain) that harbor similar mutations along these pathways might be considered as a comparator. There are clearly parallels in dysfunctional vessel morphogenesis between lymphatic and vascular malformations, though these may acquire mutations at different developmental stages of differentiation; would be valuable to mention or at least reference this comparison, similarities if the aim is to postulate potential targets for future investigation.

Minor notes:

Prox1 – please describe gene name in full at first mention and be consistent with gene/protein name labelling throughout.

Lines 109/111 – CCBE1 or CCGE1?

Line 130 – primordia not primorida

Line 149- lymphangiogenesis spelt incorrectly

Reviewer 2 Report

I am very grateful to take this opportunity to review this manuscript. This is a comprehensive review of LM and it is very well written and organized. I just have a few of minor comments for authors to consider. (1) It will be good to have a figure to summarize the current hypotheses regarding pathogenesis of LM. (2) It will be very helpful to have a histology image to show the pathological phenotype of LM. (3) Prox-1 needs to be in full name in the line 37 of page 1 but not in the line 89 of page 3.
